# Analysis of Characteristics and Clinical Outcomes for Crisis Management during the Four Waves of the COVID-19 Pandemic

**DOI:** 10.3390/ijerph182312633

**Published:** 2021-11-30

**Authors:** Watchara Amasiri, Kritsasith Warin, Karicha Mairiang, Chatchai Mingmalairak, Wararit Panichkitkosolkul, Krittin Silanun, Rachasak Somyanonthanakul, Thanaruk Theeramunkong, Surapon Nitikraipot, Siriwan Suebnukarn

**Affiliations:** 1Faculty of Engineering, Thammasat University, Pathum Thani 12121, Thailand; awatchar@engr.tu.ac.th; 2Faculty of Dentistry, Thammasat University, Pathum Thani 12121, Thailand; 3Faculty of Medicine, Thammasat University, Pathum Thani 12121, Thailand; khunpa_kiki@hotmail.com (K.M.); michatch@staff.tu.ac.th (C.M.); krittinsilanun@gmail.com (K.S.); 4Faculty of Science and Technology, Thammasat University, Pathum Thani 12121, Thailand; wararit.panichk@gmail.com; 5Sirindhorn International Institute of Technology, Thammasat University, Pathum Thani 12121, Thailand; d5922300164@g.siit.tu.ac.th (R.S.); thanaruk@siit.tu.ac.th (T.T.); 6Academy of Science, Royal Society of Thailand, Sanam Sueapa, Khet Dusit, Bangkok 10300, Thailand; 7Thammasat University Hospital, Pathum Thani 12121, Thailand; suraniti@tu.ac.th; 8Research and Innovation Division, Thammasat University, Pathum Thani 12121, Thailand; ssiriwan@tu.ac.th

**Keywords:** COVID-19, field hospital, epidemiology, risk factors, crisis management

## Abstract

This study aims to analyze the patient characteristics and factors related to clinical outcomes in the crisis management of the COVID-19 pandemic in a field hospital. We conducted retrospective analysis of patient clinical data from March 2020 to August 2021 at the first university-based field hospital in Thailand. Multivariable logistic regression models were used to evaluate the factors associated with the field hospital discharge destination. Of a total of 3685 COVID-19 patients, 53.6% were women, with the median age of 30 years. General workers accounted for 97.5% of patients, while 2.5% were healthcare workers. Most of the patients were exposed to coronavirus from the community (84.6%). At the study end point, no patients had died, 97.7% had been discharged home, and 2.3% had been transferred to designated high-level hospitals due to their condition worsening. In multivariable logistic regression analysis, older patients with one or more underlying diseases who showed symptoms of COVID-19 and whose chest X-rays showed signs of pneumonia were in a worse condition than other patients. In conclusion, the university-based field hospital has the potential to fill acute gaps and prevent public agencies from being overwhelmed during crisis events.

## 1. Introduction

The current coronavirus disease 2019 (COVID-19) pandemic is a public health emergency that requires crisis management to offer efficient tools for services and resource allocation [1]. The COVID-19 crisis has challenged governments around the world to tackle the pandemic, adopt new policies, support vulnerable communities and individuals, and find the means to achieve results under intense pressure. To deliver services to vulnerable people in confusing and difficult conditions, one key aspect is to coordinate with and be supported by other network partners, including community and non-governmental organizations [2]. University hospitals, with a large capacity in terms of human resources and settings, play an important role in crisis management. However, hospitals are overrun as another wave of COVID-19 infections fills available beds and stretches intensive care units [3]. Collaboration among university teaching hospitals, volunteers, and community groups can be considered a form of coproduction to fill acute gaps and prevent public agencies from being overwhelmed during crisis events [4]. This pandemic is a global health threat and requires collaborative action to tackle it locally and globally [5].

The number of patients with COVID-19 requiring hospitalization has increased since the first outbreak in early 2020 [6]. Due to rapid transmission, countries around the world should increase the attention paid to disease surveillance systems, scale up country readiness and establishing rapid response protocols [7]. The concept of creating a field hospital is based on the idea of focusing on patients whose condition will not require advanced treatment of coexisting diseases while relieving the burden of hospitals operating on normal principles [8]. Thailand’s first university-based field hospital project was started under a collaboration of five university hospitals to alleviate congestion at existing facilities after a jump in the number of COVID-19 patients in early 2020, when the first wave of coronavirus hit the country [9]. COVID-19 may cause rapidly worsening conditions after infection, which results in a high demand for hospital resources [10]. The field hospital is on 24 h alert to receive further admissions as hospitals spill over during this far more deadly fourth round of infections. The latest wave of cases was brought on by the delta variant of COVID-19, which is more contagious and deadly than those in the previous three waves.

The concept of creating temporary hospitals has been tested in other countries, and if the number of patients requiring hospitalization due to COVID-19 increases further, these hospitals may prove extremely useful [8]. Data on baseline characteristics and outcomes of field hospital patients with COVID-19 are essential for planning actions preceding local outbreaks and to assess the need for supportive care. Recent reports from China and the US indicated a healthy discharged rate between 81% and 86% among field-hospital patients [11,12]. Differences in patient characteristics, socioeconomic status, health care systems, field hospital admission thresholds, and the availability of field hospital beds between countries might explain such a wide difference in the outcomes.

The aim of the current study was to examine clinical characteristics and identify the risk factors related to worsening outcomes of field hospital patients with COVID-19 in Thailand’s first university-based field hospital. The experience gained during the development of a field hospital will allow for better preparation, organizing hospital resources for future units of this kind, and the more optimal use of medical personnel and equipment for managing COVID-19 patients in field hospitals.

## 2. Materials and Methods

### 2.1. Field Hospital Administration

This study was conducted at Thailand’s first university-based field hospital. The field hospital was transformed from a service apartment-style 14-story building which was previously a university dormitory into a 494-bed facility for non-critical COVID-19 patients. This field hospital is managed by the main university hospital and includes patients referred from the project’s five university hospitals and hospitals in the central area of Thailand. The sources of funding come mainly from the donations of university alumni, community groups and nongovernmental organizations.

Upon admission, a nurse records patient data in the COVID-19 screening of the field hospital information system; the patient undergoes a chest X-ray, blood tests for complete blood count (CBC), liver function tests (LFTs), and tests for electrolytes, balance urine nitrogen (BUN), and creatinine (Cr). The doctor interprets the lab tests and chest X-ray, and records the results in the admission note. The patients are only admitted to the field hospital if they meet all of the following criteria: (1) asymptomatic, mild or moderate symptoms; (2) normal activities of daily living; (3) no important organ dysfunction; (4) no psychiatric history; and (5) resting pulse oxygen saturation (SpO_2_) > 95%.

To avoid unnecessary contact between patients and medical personnel, the patient reports signs and symptoms, wants and needs via an internal field hospital application. Any consultation with the attending physician is done through a notification form. If the attending physician wishes to speak to the patient, the patient’s telephone number is obtained from the respective patient’s floor. All prescriptions must be made using a prescription form, which is then processed by the attending nurse and recorded in the progress note in the field hospital information system and in the university hospital electronic medical record system.

Laboratory and radiological examinations are performed based on the patient’s history of taking favipiravir. For favipiravir-naive patients: (1) a follow-up chest X-ray may be considered in patients with worsening signs and symptoms (body temperature (BT) > 38.0 °C, coughing, fatigue, SpO_2_ < 96%, or decreased SpO_2_ > 3% after a stress test); and (2) if the chest X-ray infers pneumonia with respiratory signs and symptoms (as mentioned in (1), the patient is referred to the originating hospital for continued treatment with favipiravir. For patients previously treated with favipiravir: (1) follow-up by chest X-ray and LFTs is performed; (2) if LFTs increase, an ID specialist may be consulted to terminate/adjust medication use; and (3) if the chest X-ray infers a progression of the infiltration accompanied by respiratory signs and symptoms (cough, fatigue, SpO_2_ < 96% and SpO_2_ drop < 3% after a stress test), the patient may be referred to the hospital of origin.

Asymptomatic patients who have been hospitalized for at least 14 days after a positive COVID-19 test will be discharged home. The patients who received favipiravir should fulfil all of the following criteria: (1) the patient’s signs and symptoms have improved without the progression of infiltration on chest X-ray; (2) BT < 37.8 °C continuously for 24–48 h; (3) respiratory rate (RR) < 20/min; and SpO_2_ > 96% at rest. In the event of a patient’s condition deteriorating, they are quickly transferred to the designated higher-level hospitals. The criteria for transfer are (1) meeting the criterion of severe or critical, and (2) lung imaging showing a greater than 50% progression of lesions.

Patients do not need real-time polymerase chain reaction (RT-PCR) or antigen/antibody detection for COVID-19 prior to discharge. One day before discharge, the attending nurse informs the attending physician of the number of potential discharges, so that the physician can prepare medical certificates and insurance documents according to the patient’s needs. Upon discharge, the attending physician updates the patient’s progress and discharge summary in the electronic medical record system of the university hospital.

### 2.2. Data Collection

Registry data were retrieved from the electronic hospital information systems of the referral hospitals and the field hospital information system. In this study, we included all patients confirmed with asymptomatic and mild-to-moderate COVID-19 symptoms from March 2020 to August 2021 (covering four waves of COVID-19 in Thailand) (Figure 1). The collected data included patient demographics, comorbidities, body mass index (BMI), job, place of exposure to coronavirus, symptoms before field hospital admission, signs of pneumonia in the chest X-ray and field hospital length of stay. The outcome measure was the field hospital discharge destination.

### 2.3. Statistical Analysis

Categorical variables are presented as numbers (with percentages). Continuous variables are summarized as medians with the interquartile range (IQR). Data analyses were conducted using the Statistical Package for the Social Sciences version 22.0 (SPSS, Chicago, IL, USA). Data were presented in categories for age (<44 years; 45–64 years; and ≥65 years), sex (male; and female), body mass index (BMI) (<25 kg/m^2^; 25–29 kg/m^2^; and ≥30 kg/m^2^), comorbidity (no comorbidity; respiratory disease; hypertension; dyslipidemia; metabolic disease; or others), job (general worker or healthcare worker), symptoms before field hospital admission (asymptomatic; mild; and moderate), signs of pneumonia in the chest X-ray (no lesion; pneumonia), place of exposure to coronavirus (family; community; and hospital or clinic), field hospital length of stay (≤14 days; and >14 days), and field hospital discharge destination (discharged home or transfer to designated high-level hospitals due to condition worsening).

Assessment of the following potential risk factors for field hospital discharge destination was performed using multivariable logistic regression analysis: age, sex, BMI, comorbidities, job, symptoms before field hospital admission, signs of pneumonia in the chest X-ray and place of exposure to coronavirus. A *p* value < 0.05 was considered statistically significant.

## 3. Results

A total of 3685 patients (53.6% female; median age 30 (IQR 12–48) years), including patients with asymptomatic and mild to moderate COVID-19 were admitted to our field hospital between March 2020 and August 2021 (Figure 1). The median BMI was 23.3 (IQR 16.4–30.2) kg/m^2^. Overall, 3392 (92.0%) patients had no known comorbidity. The most prevalent comorbidities included respiratory disease (2.2%), including asthma and allergic rhinitis, and metabolic syndrome (1.4%). A total of 3592 (97.5%) patients were general workers. Among the 93 (2.5%) other patients, 13 (0.35%) were physicians, 17 (0.46%) were nurses, 1 (0.02%) was a medical technologist, 2 (0.05%) were pharmacologists, 3 (0.08%) were medical students, 3 (0.08%) were nursing students and 54 (1.46%) were general healthcare workers. According to the patient’s timeline information, the patients were exposed to coronavirus from the community (84.6%), family (12.9%) and hospital or clinic (2.5%). Most of the patients, 2295 (62.3%), had no signs and symptoms, 1371 (37.2) had mild symptom and 19 (0.5) had moderate symptom upon admission. As a standard guideline, the patients needed to be hospitalized at least 14 days after a positive COVID-19 test. In this study, the patients had been hospitalized prior to being referred to our field hospital. At the study end point (22 July 2021), 3625 (98.4%) stayed no more than 14 days at the field hospital, no patients died, 3600 (97.7%) were discharged home and 85 (2.3%) were discharged to hospital wards (Table 1).

In the multivariable logistic regression analysis, older age (OR 1.019 per year, 95% CI 1.003–1.036, *p* < 0.001), having one or more underlying diseases (OR 1.218, 95% CI 1.052–1.410, *p* < 0.01), mild or moderate symptoms prior to admission (OR 2.977, 95% CI 1.890–4.691, *p* < 0.01) and sign of pneumonia in chest X-ray (OR 0.182, 95% CI 0.078–0.424, *p* < 0.001) were significantly associated with worsened conditions that required transfer to designated high-level hospitals. We observed no independent association of sex, BMI, place of exposure to coronavirus and job with field hospital discharge condition (Table 2).

## 4. Discussion

This study provided comprehensive COVID-19 patient data on epidemiologies, demographics, clinical characteristics and patient conditions prior to admission as well as the outcomes of hospitalization at our field hospital during four waves of the COVID-19 pandemic. In our study, most of the patients were under 44 years old, and were predominantly female. Most of the patients had no underlying disease. These results are different from previous studies in China, Italy and the United States of America, where most of the patients were over 65 years of age, predominantly male, and had one or more underlying diseases [6,13,14]. Most of the patients in this study were identified as asymptomatic and mild symptom cases, in accordance with previous studies on this novel severe acute respiratory syndrome coronavirus-2 (SARS-CoV-2), which frequently causes only mild symptoms similar to the common cold, and asymptomatic carriers [15,16].

In this study, we identified the place where patients contracted SARS-CoV-2 virus and found that most of the patients were infected from community sources such as markets, bars and entertainment venues. The second largest group was infected through contact with family members who were previously confirmed with COVID-19 infection. The last significant cause, which should not be overlooked, was nosocomial infection from hospitals and private clinics. As we know, COVID-19 spreads via airborne transmission [17], and can easily infect those in close contact with infected patients. This may make healthcare workers more susceptible to be infected with the SARS-CoV-2 virus. Therefore, we were interested in identifying the proportion of healthcare workers who were infected with COVID-19, and found that 2.5% of patients in our field hospital were healthcare workers. The result was similar to previous studies, which showed that healthcare workers were at high risk of COVID-19 infection and at higher risk with long working hours [18]. Most of the healthcare workers infected with COVID-19 were physicians and nurses. This confirms that healthcare workers who are in close contact with patients may have a higher risk of infection. In particular, physicians who perform procedures around the head and neck area, such as otolaryngologists, oral and maxillofacial surgeons, plastic surgeons, anesthesiologists and dentists, may experience more exposure to aerosol-generating procedures and thus may be more susceptible to infection with this airborne virus.

We evaluated the demographic and clinical risk factors on worsening conditions that required transfer to designated high-level hospitals in our field hospital, and found that older age, having one or more underlying diseases, mild or moderate symptoms prior to admission and signs of pneumonia in the chest X-ray were significantly associated with worsening conditions. The results are similar to previous studies which found that higher age, one or more comorbidities, and being overweight were significantly related to severe conditions and death in COVID-19 patients [13,19,20,21].

The studied field hospital is Thailand’s first university-based field hospital with a service apartment style, supporting almost 4000 patients over the course of the four COVID-19 pandemic outbreaks in Thailand since April 2020. For COVID-19 patient safety, we designed admission criteria only for asymptomatic and mild symptom cases. However, for an urgent situation with a limit on university hospital resources, we also admitted cases with moderate symptoms with the close monitoring of patients. The care processes with 24 h alert are provided to the patient’s changing condition with on-site medical care or by transporting the seriously ill patient to the university hospital. This system has allowed us to treat 97.7% of our patients in place, with successful discharge and no cardiac arrests or on-site deaths since opening. Currently, Thailand is facing a renewed crisis in the COVID-19 pandemic, with a large number of new cases, at approximately 20,000 cases per day [22]. With the high volume of COVID-19 patients, especially critically ill patients, rapidly exceeding the capacity of local and general hospitals, the development of a field hospital was expected to help increase the capacity to deal with the surge in COVID-19 patients, organize hospital resources and the distribution of hospital capacity for asymptomatic patients and patients with mild or moderate symptoms. Currently, there are studies on the development of field hospitals in many countries, especially countries with a large number of COVID-19 patients and limited health care capacity, including China, Italy and the United States of America. These studies aimed to create health systems to establish efficient and effective medication distribution services to help the health care workflow and support patients in this crisis [11,23,24].

The results of the data analysis and the experience gained during the development of our field hospital led us to further develop a crisis management protocol (home isolation) and a disease control program (a vaccination center). The main objective of home isolation was to make optimal use of limited medical resources during the surge in COVID-19 patients. The home isolation protocol was a telemedicine-based COVID-19 surveillance system. COVID-19 patients were screened and classified into two groups of asymptomatic and symptomatic patients. Only symptomatic patients were admitted to the field hospital according to a protocol explained in this article. COVID-19 patients who enlisted for home isolation were be counseled, prescribed COVID-19 drug sets, and monitored for the progression of COVID-19 symptoms, including body temperature and SpO_2_, at home by volunteer physicians via telemedicine twice a day for 14 days. All adverse conditions were notified and reported, and patients were immediately transferred to the university hospital. In addition, to prevent the spread of the disease, the university vaccination center was established as a COVID-19 control program, which is expected to reduce the influx of COVID-19 patients so as not to exceed the capacity of the hospital.

There are limitations to this study. First, the sample of this study was limited only to the central area of Thailand. Second, some data, including the co-morbidities, were missing, along with varieties and severity of the disease due to the incomplete recording system in the early wave of COVID-19 pandemic in Thailand. For the future improvements of our field hospital system, we plan to incorporate artificial intelligent data mining technology to provide medical alerts from the data recording, motion and object detections for infection control, and the automated security system for real-time monitoring of patient safety in the field hospital.

## 5. Conclusions

This study of patients admitted to Thailand’s first university-based field hospital presents baseline characteristics and clinical course, with 0% mortality and 98.8% being discharged home. Older age, having one or more underlying diseases, symptomatic cases and signs of pneumonia in the chest X-ray were associated with higher worsening conditions that required transfer to designated high-level hospitals. The crisis management protocol and a disease control program developed based on the results of the data analysis led to better preparedness, the organization of hospital resources and more optimal use of personnel and medical equipment to manage COVID-19 in this crisis situation and should be the preparation protocol for the next wave of COVID-19.

## Figures and Tables

**Figure 1 ijerph-18-12633-f001:**
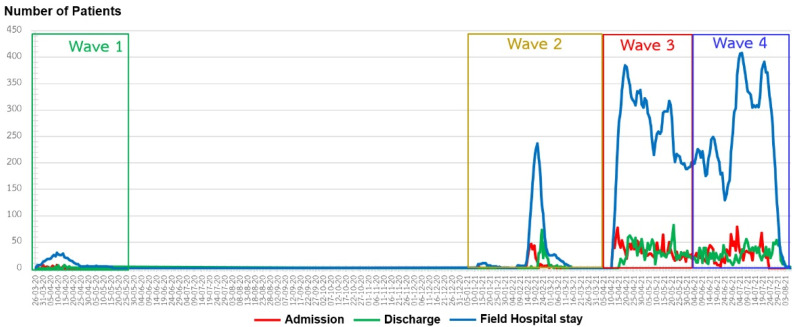
Daily number of new field hospital admission with COVID-19, field hospital length of stay, and discharge from March 2020 to August 2021.

**Table 1 ijerph-18-12633-t001:** Baseline characteristics.

Characteristic	All Patients (*n* = 1931)
**Age (year)**	30 (12–48)
0–44	2986 of 3685 (81.0%)
45–64	625 of 3685 (17.0%)
>65	74 of 3685 (2.0%)
Sex	
Male	1711 of 3685 (46.4%)
Female	1974 of 3685 (53.6%)
**Body mass index (kg/m^2^)**	23.3 (16.4–30.2)
<25	2309 of 3685 (62.7%)
25–29	931 of 3685 (25.3%)
≥30	445 of 3685 (12.0%)
**Comorbidity**	
None	3392 of 3685 (92.0%)
Respiratory disease	82 of 3685 (2.2%)
- Asthma	
- Allergic rhinitis	
Hypertension	39 of 3685 (1.1%)
Diabetes mellitus	18 of 3685 (0.5%)
Dyslipidemia	14 of 3685 (0.4%)
Metabolic syndrome	53 of 3685 (1.4%)
Pregnancy	23 of 3685 (0.6%)
Others	64 of 3685 (1.7%)
- Thalassemia	
- Thyroid disease	
- Gout	
- G6PD deficiency	
**Job**	
General worker	3592 of 3685 (97.5%)
Healthcare worker	93 of 3685 (2.5%)
Physician	13 of 3685 (0.35%)
Nurse	17 of 3685 (0.46%)
Medical technologist	1 of 3685 (0.02%)
Pharmacologist	2 of 3685 (0.05%)
Medical student	3 of 3685 (0.08%)
Nurse student	3 of 3685 (0.08%)
General healthcare worker	54 of 3685 (1.46%)
**Place of exposure to coronavirus**	
Community	3119 of 3685 (84.6%)
Family	475 of 3685 (12.9%)
Hospital or clinic	91 of 3685 (2.5%)
**Symptom before field hospital admission**	
Asymptomatic	2295 of 3685 (62.3%)
Mild	1371 of 3685 (37.2%)
Moderate	19 of 3685 (0.5%)
**Chest X-ray**	
No lesion	3213 of 3685 (87.2%)
Sign of pneumonia	472 of 3685 (12.8%)
**Field hospital length of stay**	
≤14 days	3625 of 3685 (98.4%)
>14 days	60 of 3685 (1.6%)
**Field hospital discharge destination**	
Discharged home	3600 of 3685 (97.7%)
Transfer to designated high-level hospitals due to condition worsen	85 of 3685 (2.3%)

G6PD, glucose-6-phosphate dehydrogenase.

**Table 2 ijerph-18-12633-t002:** Multivariable logistic regression analysis showing the association between different variables and worsening conditions requiring transfer to designated high-level hospitals.

Variables	Odds Ratio (95% CI)	*p* Value (<0.05)
Older age	1.019 (1.003–1.036)	0.020 *
One or more underlying diseases	1.218 (1.052–1.410)	0.009 *
Mild or moderate symptom	2.977 (1.890–4.691)	0.000 *
Sign of pneumonia in chest X-ray	0.182 (0.078–0.424)	0.000 *

* Significant at *p* Value < 0.05.

## Data Availability

The data are not publicly available due to privacy reasons.

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
