# Peer review of "Analysis of Characteristics and Clinical Outcomes for Crisis Management during the Four Waves of the COVID-19 Pandemic"

_ijerph, 2021, doi:10.3390/ijerph182312633_

Round 1

Reviewer 1 Report

Dear Authors
The theme you present is very interesting and pertinent. The studies on COVID-19 are fundamental for us to understand what happened and prepare ourselves for the future. 
In general your article is well written and interesting, but you should improve the following:
1) the article will be more interesting if you introduce a Literature Review topic;
2) The conclusions of such an interesting study cannot be summarized in a few lines of text. They should be more complete;
3) Review the references.

After these changes the article will be more complete and able to be published
Best Regards

Reviewer 2 Report

The analysis of the characteristics of patients admitted to the field hospital along with the evaluation of their health condition provides the basis for better organization of health care, already burdened by the ongoing epidemic;
An epidemic is a kind of cataclysm, and in mass incidents we triage patients. Therefore, I consider the study valuable and helpful in planning the organization of health care in times of crisis.
Nevertheless, I recommend that you amend the manuscript.
There are too many patient characteristics - ok, it should be because it implies further actions, but in the paragraph on the purpose of the study we read "The experience gained during the development of a field hospital will allow for better preparation, organizing resources of further such units and more optimal hospital. use of medical personnel and equipment for managing COVID-19 patients in the field hospital. "
Please refer to this issue; To what extent has this objective been achieved? How and to what extent can this research contribute to better organization of healthcare? Only the discussion of this topic would be really valuable in this work; the subject of the work promises a lot but in my opinion the results and the discussion do not reflect the given issue;
Additionally, please change the discussion - too much data strictly repeated from the results of the work, too little discussion.

Round 2

Reviewer 2 Report

Dear Authors,

The introduced corrections will certainly strengthen the presented report. I will advise you to avoid repetitions from the Results section in the Discussion section.

Author Response

The introduced corrections will certainly strengthen the presented report. I will advise you to avoid repetitions from the Results section in the Discussion section.

Response : We thank the reviewer for the comment. We have removed the repetitions from the Result section in the Discussion section (in paragraphs 1, 2, 3), leaving only the details for discussion with related studies.